# Novel Approaches Needed: An Experimental Study with an Alternative to Mechanical Restraint

**DOI:** 10.3390/healthcare12161658

**Published:** 2024-08-20

**Authors:** Tilman Steinert, Bernd Maierhofer, Peter Schmid, Sophie Hirsch

**Affiliations:** 1Klinik für Psychiatrie und Psychotherapie I, Ulm University (Weissenau), Weingartshofer Str. 2, 88214 Ravensburg, Germany; 2Centres for Psychiatry Suedwuerttemberg, Weingartshofer Str. 2, 88214 Ravensburg, Germany; bernd.maierhofer@zfp-zentrum.de (B.M.); peter.schmid@zfp-zentrum.de (P.S.); 3Centres for Psychiatry Suedwuerttemberg, Psychiatry Biberach, Paracelsusweg 3, 88400 Biberach, Germany; sophie.hirsch@zfp-zentrum.de

**Keywords:** restraint, alternative, coercion, milder means, human dignity

## Abstract

Despite many calls to reduce or eliminate the use of mechanical restraint, it is still widely used in many countries. Studies using patient interviews have a very clear message: Patients experience mechanical restraint as the most humiliating intervention. There seems to be a lack of alternatives for violent patients if all other approaches to prevent the use of coercion have failed. We developed a method using 30 kg bags, originally designed for fitness purposes, to be attached to a patient’s wrist or ankle under 1:1 supervision. The method was tested with 10 experienced nurses and de-escalation trainers. A video was made and presented to six outpatients who had previously experienced mechanical restraint. All participants were interviewed. Transcribed interviews were analysed using qualitative content analysis. All participants approved of the method as a milder and less humiliating alternative to mechanical restraint. The nurses’ main concerns were the risk of falls and the use of the bags as weapons. The latter could be controlled by using an additional bag. Patients were generally positive, especially if there was a history of abuse. The method should be further developed to replace at least some mechanical restraints. As with all ‘milder means’, care should be taken to really replace restraint and not to introduce additional coercion.

## 1. Introduction

Mechanical restraint is used in most countries to control aggressive and violent inpatients. Mechanical restraint means restraining patients by attaching belts to their bed at four (wrists, ankles), five (including abdomen), or even seven points (including head and shoulders) to achieve immobilisation and avoid harm to themselves and others. Surveys suggest that it is used in between 3% and 8% of all psychiatric admissions [1,2], amounting to millions of cases worldwide each year. The only alternatives are seclusion and, to some extent, the use of compulsory medication. Seclusion means that the patient is placed in a locked room with very little furniture for security reasons, often just a mattress on the floor. Staff remain outside the seclusion room, maintaining contact through a window and sometimes watching on a video screen. A few countries—the United Kingdom, Ireland, the Netherlands, and Switzerland—have a clear preference for seclusion, with mechanical restraint used only in very exceptional cases [1,3]. Interviews with patients from different countries gave very clear results: Patients find mechanical restraint extremely humiliating, and most (but not all) patients find it more intrusive and distressing than any other measure [4,5,6,7,8,9]. However, seclusion is also perceived as humiliating; staff contact and medical checks such as blood pressure measurement are difficult to realise during seclusion [4,6]. In Denmark, for example, seclusion is banned because it is considered inhumane [10]. The use of tranquilising medications is limited for medical reasons, and their use is also controversial in many countries [3,11]. The UN Committee on the Convention on the Rights of Persons with Disabilities and the World Health Organisation strongly recommend that all countries abolish these practices and the corresponding domestic legislation [12,13], without, however, providing reasonable alternatives for people who are violent because of their mental disorder. In fact, convincing examples of a practice that eliminates seclusion and restraint have not yet been demonstrated in any country, and long-term data do not show a significant reduction [2]. The main reason is violence, observed in a significant proportion of psychiatric admissions [14,15], which has never been discussed in the appeals to eliminate the use of coercion. If a sanitarian is attacked, the police are called to bring the guy away. If a nurse is attacked in a psychiatric ward, the respective patient will stay there. Therefore, it does not seem justified to wait until mechanical restraint is abolished. Given the undoubtedly humiliating nature of mechanical restraint, it can be considered a shame that, in the 21st century, we have not developed less humiliating and safer alternatives. There is an urgent need for novel approaches. Here, we describe a pilot study of one such approach.

## 2. Methods

### 2.1. Technique

We developed a method with 30 kg bags, originally developed for purposes of fitness training, that can be attached with a length-adjustable belt at its handle to a patient’s wrist or ankle. This allows patients to sit, eat, or go to the toilet. In cases of severe agitation and the danger of attacking the nurse present in the room, a second bag can be attached to the first bag in a row or at another wrist or ankle. A 4 min video demonstration with subtitles in English is available at: https://www.uniklinik-ulm.de/fileadmin/default/Kliniken/Psychiatrie-Psychotherapie-I/Bilder/2023-05-03_Fixierung_480_UT1_6mbps_str.mp4 (accessed on 19 August 2024).

### 2.2. Experimental Study

As long as these bags are not yet certified medical devices, they cannot be used with patients, even on a voluntary basis. Furthermore, any alternative to mechanical restraint or seclusion can only be tested in real-life conditions on involuntary patients without informed consent. Otherwise, the procedure would not be coercive and would not represent real clinical conditions. No ethics committee would approve such an intervention without extensive prior work. Therefore, we conducted an experimental study, tested the method with volunteer professionals, and presented the video demonstration to patients who had previously experienced mechanical restraint. The aim of the pilot study was to explore potential risks and side effects, as well as scenarios for appropriate use, and to receive an ethical assessment compared to mechanical restraint.

### 2.3. Test Scenario

Participants underwent experimental mechanical restraint in a bed with belts at five points. Then restraint belts were removed, and, as an alternative, a 30 kg bag was attached to their wrist. Then, they were encouraged to move, eat, go to the toilet, and test the possibility of a violent attack on the other nurse present in the room. This nurse was instructed to try to reach a safe distance in the event of an attack. In this situation, the use of a second bag (in a row with the other, at the other wrist, or at an ankle) was also tried.

### 2.4. Ethics

The ethical thresholds to test a coercive intervention on involuntary patients are very high. We are well aware that no ethical board would approve mechanical restraint as a new intervention if it were not common practice for centuries. However, for any novel intervention, evidence of ethical merit and adequate safety is required before real-world trials can begin. We chose to design an experimental study on ethical grounds, avoiding the direct exposure of patients to a coercive intervention. A positive ethical vote was obtained from the Medical Chamber of Baden-Wuerttemberg (F-2023-016). All participants received written study information and signed informed consent for their participation, including recording the interview and using the data.

### 2.5. Participants

Professional participants were recruited from the centres of psychiatry in Suedwuerttemberg, a hospital group with 10 sites in the Federal State of Baden-Wuerttemberg. Inclusion criteria were ≥three years of work in acute psychiatry, experience with using mechanical restraint, and experience with de-escalation training.

The inclusion criterion for patients was having experienced at least one mechanical restraint. Being an inpatient was an exclusion criterion. Patients were recruited among the centres’ outpatient services and self-help groups.

### 2.6. Interviews

All interviews were conducted by TS using a previously developed interview guideline. First, the scenario was clarified: The bag technique should replace the mechanical restraint conducted before, with a nurse 1:1 present continuously and nobody else in the room. Then, participants were asked which risks for patients they could imagine under these conditions and which risks for staff. Subsequently, they were asked to imagine the method for different types of patients and to judge whether this could be appropriate: Agitated patients with psychotic disorders or mania, violent patients with delirium and dementia, intoxicated patients, patients with mental retardation and violent behaviour towards themselves or others, and patients with borderline personality disorder and repeating severe self-harm. Finally, participants were encouraged to express their ethical opinions in terms of human dignity, fairness, and humiliation. Interviews were recorded and then transcribed.

### 2.7. Analysis

All interviews were analysed by TS and SH separately with qualitative content analysis [16] according to themes. No ‘vote counting’ was done; instead, we attempted to represent the diversity of opinions. Quotations are presented for typical opinions.

## 3. Results

The professional participants were eight male and two female nurses. Two worked as de-escalation trainers and eight as nurses in acute psychiatric wards. Their median age was 45 years (34–59), and they had a median of 20 years of professional experience (4–33). The patients were four females and two males, with a median age of 40 years (30–55). They had a median of 15 psychiatric admissions (4-estimated 100) and had experienced a median of 2 mechanical restraints (1-estimated 100). Three patients suffered from schizophrenia, two from bipolar disorder, and one from complex posttraumatic stress disorder.

### 3.1. Risks for Patients

According to the professionals, falls were assumed to be the major risk of the technique. “Because at that moment, he might trip over his own legs because he’s dangerous to others; you are a few steps away, and I cannot get there that quickly”. This risk was also rated as high because the patients could not easily catch themselves with the bags on their hands in the event of a fall: “I see the risk that he could injure himself and then possibly hit his face head-on, a head injury”. As a minor risk, some professionals imagined patients could hurt their shoulders or their back with abrupt movements. “I could imagine it, but I would be worried that he forgets that something is attached to him and gets up suddenly and wants to run after me, for example, and that something happens with the shoulder or the patient falls down” or that the belts could cut in. One nurse worried that the bag could fall on the patient “because it can also be dangerous if he drops something on his foot from these weight bags”. Further comments referred to the material: Seams must be stable and robust, and aspects of hygiene have to be considered (washable material, etc.). Some professionals expressed concerns that patients with a risk of severe self-harm could strangle themselves with long belts but conceded that this risk could be averted by the presence of a nurse in 1:1 supervision.

One of the patients feared that self-harm could not be prevented as safely with this technique as with others. Another was concerned that the belts could cut the skin. Overall, however, fewer concerns were expressed by the patients than by the staff.

### 3.2. Risk for Staff

All the interviewed professionals and some of the patients saw the major risk of staff in the bags being used as weapons, attacking the nurse present in the room. Some said that they would not use it for very strong and agitated patients for this reason. However, all conceded that this risk could be minimised by the use of a second bag in a row or at the other wrist. They mentioned that this would then be rather similar to traditional mechanical restraint. The somewhat surprising experience for participants was that they were well able to move but rather slowed down and soon exhausted. “Of course, the radius is such that you could be hit, but you can always keep a safe distance if you work a bit ahead”. “It is about the speed of the patient; you are out of the area so quickly where you would be at risk”. Staff expressed the concern that patients could kick with their legs, as these are not restrained in contrast to conventional restraints: “Right after overpowering him, I am just a bit worried that the legs are free and that someone else could get hurt”. Another concern of some participants was the weight of the devices for transportation. This could hurt the back, especially among those with lower body weight. There were controversial opinions on the appropriate shape of the bags. Good handles would be useful for transportation but could also facilitate violent acts by patients. The form of a cone instead of a bag would enhance friction on the ground and help to slow down the patient, but it would be more difficult for transportation. Less or more weight in the bag could be more appropriate for individual patients, but the aspect of transportation should be taken into account. “Of course, it should not be so heavy that it makes movement completely impossible, but at the same time, it should have enough weight so that the patient has difficulty, for example, in hurling it or somehow being able to misuse it so easily”.

Patients only addressed the aspect of possible violent acts against staff.

### 3.3. Indications

The professional participants agreed that they would not apply the method to elderly patients with a risk of falls or severely intoxicated patients due to the same reasons. “I could well imagine to use it with aggressive patients with psychosis or mania” was a frequent comment. However, there were also concerns expressed relating to very aggressive, agitated, and vigorous patients. For patients with mental retardation and aggressive behaviour towards themselves and others, opinions were inconsistent. Some participants were sceptical, others positive: “This method would perhaps also be an addition to the spectrum in this respect; I could well imagine that”. With regard to patients with severe self-harm, different views were expressed. Some saw no use and supposed that such patients would try to strangle themselves or bang their heads on the ground. Others expressed more positive opinions: “But to relieve tension, those who inflict pain and wounds on themselves would have to weigh it up; yes, that would perhaps also be something you could discuss with the patient”. Several participants said that introducing a new method would require new standards of quality to be established: “Yes, I think you have to discuss with the team beforehand which patient would be suitable”. Many professionals stated that they could well imagine the bags as a loosening or interim solution after the end of restraint or seclusion in order to reduce its duration.

Interviewed patients were less able to differentiate different types of patients from their own experience. They expressed fewer concerns with possible violence: “I really believe that those who are so excited, so agitated, that they then lash out with 30 kg bags—that is a vanishingly small minority. So I think so; I can well imagine it with most of them”. One patient expressed a strong opinion that the method should be absolutely preferable for those who had been victims of abuse: “So I speak for all victims of abuse, and I do not think there is an exception, that this restraint on the back is the most terrible situation…”.

### 3.4. Human Dignity

Without exception, professionals stated that they found the novel approach less humiliating and ethically preferable for those to whom the procedure could be applied without undue risk to themselves or staff. “One has the association that it is a bit like the iron ball on the foot and reminds one of the convict camp. But actually, mechanical restraint is much more restrictive and humiliating. In this respect, it is a milder means in comparison. I think it is very good that people are thinking about such alternatives and would like to try them out if it were possible”. “Coercion is coercion, but if I imagine many of our patients are secluded, restrained with belts, and then have to pee and go to the toilet, they could do that with it, and I think that is a point that convinces many. The need for a bedpan is undignified for all”. The nurses interviewed had the idea that the measure could be helpful in returning people to the community on the ward but were concerned that they could be stigmatised or ridiculed if they wore the bags: “At first, I thought that the idea might be that the patient could sit at the table properly outside the seclusion room, for example, that he could eat. […] However, I see the risk of stigmatisation if patients see it that way”.

Patients interviewed were very positive about an alternative and often referred to their own negative experiences with mechanical restraint: “That is why I find this alternative a thousand times better from my point of view, that I would not have these choking attacks because I could sit and would not have to lie on my back”. Also, psychotic experiences during restraint played a role: “Exactly, you are not tied down on a bed and then facing the lethal injection or something. That is much more pleasant here. Because that is another association”. Negative associations like an “iron ball” were not expressed. Compared to seclusion, the possibility of contact was emphasised: “Yes, it is a good thing because you can stay in touch”. Patients also mentioned the aspects of eating and going to the toilet: “So you could still eat yourself and, if necessary, also go to the toilet yourself, so you feel completely inhumane when you are fed, or things like that are done with you”.

## 4. Discussion

The findings of our experimental study show that this new approach could be promising. Strictly speaking, it is also a mechanical restriction of mobility and, therefore, a kind of mechanical restraint. However, compared to traditional mechanical restraint, it is more flexible, allowing the patient more mobility and allowing them to use the toilet to eat and drink in a seated position and to talk to a present staff member in an upright position. In terms of human dignity, all participants rated it positively compared to mechanical restraint. No undue risks or serious safety concerns were expressed. The main safety concern was whether a 30 kg bag would be sufficient to control a strong, agitated, and violent patient. However, all participants agreed that the bags are a flexible tool that can be fine-tuned to individual situations. For weaker patients, a 20 kg bag would probably be sufficient and allow more mobility. For very strong patients, two 30 kg bags in a row will be sufficient to prevent any fast movements and attacks out of reach of the free hand. Participants were divided on whether the technique could replace a small or large proportion of traditional mechanical restraints, with the main concern remaining serious violence against accompanying carers. This is not surprising given the lack of clinical experience.

It is necessary to place this proposed alternative to mechanical restraint in context. There is a huge amount of literature on ‘alternatives’ to mechanical restraint. Most of these ‘alternatives’ are interventions to prevent and avoid mechanical restraint, which are undoubtedly necessary and effective. We have previously published a systematic review of this type of intervention [17]. In a subsequent scoping review on alternatives to the use of mechanical restraints in the management of agitation or aggression in psychiatric patients [18], the authors found more or less the same studies and approaches but now call them ‘alternatives,’ following the political discussion to develop alternatives to the use of seclusion and eventually to abolish the use of coercion in psychiatry [12,13]. The ‘alternatives’ identified in the 21 included articles were staff training in de-escalation techniques, risk assessment tools, data monitoring, patient involvement, appropriate physical environments, organisational change, and complex programmes such as the Six Core Strategies or Safewards. It is also not surprising that patients least appreciate mechanical restraints and prefer ‘soft’ interventions such as listening to music, being accompanied in 1:1 supervision, soft rooms, etc. [19,20]. However, we believe that it blurs the boundaries and is like comparing apples and oranges to call these important approaches ‘alternatives.’ Fire prevention is important, but it is not a cure for burns. Similarly, an alternative to mechanical restraint is a practice that can be used when all preventive approaches have failed and a patient is still imminently or openly violent. In these situations, alternatives are very limited. The most commonly used restraint is seclusion, with considerable backrest to limit contact with the patient and to make medical assessments, such as blood pressure monitoring, more difficult. Subjective distress during seclusion is only slightly less than during mechanical restraint [4,6,21]. Physical restraint, i.e., holding the patient by physical force, is a practice predominantly used in the UK and Ireland, where mechanical restraint is not permitted [1]. It can be considered a good alternative, but it requires extensive staff training and carries risks for staff and patients [22]. To our knowledge, physical restraint could not be implemented broadly as an alternative to mechanical restraint in any of the countries where mechanical restraint is permitted. Another recently developed alternative is the so-called restraint chair, which is basically a technical variant of the traditional belt-to-bed mechanical restraint. A first qualitative study reported positive patient experiences [23]. Whether this could be a humane advance, whether there are advantages over traditional mechanical restraint, and whether there are safety issues remain to be investigated. Some further interesting alternatives to mechanical restraint have been developed in geriatric psychiatry, where there is evidence that mechanical restraint is harmful and can lead to serious somatic consequences. Although not supported by strong evidence, technical devices to prevent falls or mitigate their consequences, such as hip protectors [24,25,26], low-low beds [27], bedchair pressure sensors [28], gait stabilisers [29], and physical training [30,31], may be alternatives to restrictive interventions. However, these alternatives to mechanical restraint can only be considered in frail patients, not in those with overt violent behaviour.

Therefore, we believe that our proposed approach could be an alternative to the use of mechanical restraint for patients with overt or imminent violence when all other measures to prevent violence and coercion in inpatient psychiatric settings have failed.

The next steps in gaining such further experience with the technique, always accompanied by 1:1 staff supervision, should be similar to the introduction of new drugs. A certified medical device must be developed for any clinical use. National legislation regarding the use of coercive measures must be taken into account. Well-designed randomised controlled trials should be conducted before any introduction into the clinical routine. These trials must be designed to ensure that the novel approach does, in fact, replace mechanical restraint and does not result in additional restraint for patients who would not have been restrained in typical clinical practice.

## 5. Conclusions

As mechanical restraint is widely used in clinical practice worldwide and there is a strong call for milder alternatives, the impact on clinical practice in mental health care could be significant. The surprisingly positive comments from former patients underline the need to test this approach in real-life practice. An obstacle to take into account, at least in some countries, is the need to obtain a label as a licensed medical product. We caution against uncontrolled use and emphasise the need for rigorous studies and the development of practice guidelines.

## Data Availability

The original contributions presented in the study are included in the article; further enquiries can be directed to the corresponding author.

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
