# Peer review of "Novel Approaches Needed: An Experimental Study with an Alternative to Mechanical Restraint"

_healthcare, 2024, doi:10.3390/healthcare12161658_

Round 1

Reviewer 1 Report

Comments and Suggestions for Authors

The manuscript named “Novel Approaches Needed: A Suggestion for an Alternative to Mechanical Restraint in the 21st Century” is an interesting and meaningful study in care practice for patients with mental and psychological problems. But there are some problems. 

As for the title, this article is a review article, but in reality, it is an exploratory pilot study, so the title should be changed: A designed bag for an alternative to mechanical restraint: a pilot study. 

Since this is a pilot study, limited sample is existing. In this study, the inclusion criterion for patients was having experienced at least one mechanical restraint which means all of participants were in relative normal status during observation, which did not consider different types of disorder patients, which may greatly limit the feasibility exploration of this method.   

Although in terms of human dignity in this study, all participants rated it positively compared to mechanical restraint, there are some potential problems in Heavy bag: for transfer from one place to another; for robust, overt violent behaviour and agitating patients, a 30kg bag is not enough to secure them well, while for petite patients, a 30kg bag is a big burden; in addition, bag may fall on the patient; and other potential dangers, which may limit its application into clinic settings. 

Page 2 line 86 and 87: “experience” should be changed to experienced or experiencing.

Page 4 line 145: please cross out  in front of “Its”. 

Comments on the Quality of English Language

I think the English expression in the article is standardized and accurate, but some details still need to be modified.

Author Response

Dear Reviewer 1,

Comment: The manuscript named “Novel Approaches Needed: A Suggestion for an Alternative to Mechanical Restraint in the 21st Century” is an interesting and meaningful study in care practice for patients with mental and psychological problems. But there are some problems.

Author’s reply: Thank you for the comments that help to improve the manuscript.

Comment: As for the title, this article is a review article, but in reality, it is an exploratory pilot study, so the title should be changed: A designed bag for an alternative to mechanical restraint: a pilot study.

Author’s reply:

Comment: This is not a review article, of course. The assignment to this category is misleading. However, it is also not a pilot study. A pilot study involves a small number of subjects to the planned intervention to test its feasibility. We did not test the intervention with patients, as this would have been necessarily on an involuntary basis and no ethical board would have approved this procedure. We applied the interventions with volunteering professionals and showed the video to former patients, so it should be rather called an experimental study. We changed the title accordingly into Novel Approaches Needed: An experimental Study with an Alternative to Mechanical Restraint.  

Furthermore, we added the following in the method section (under headline experimental study): “Furthermore, any alternative to mechanical restraint or seclusion can only be tested in real-life conditions on involuntary patients without informed consent. Otherwise, the procedure would not be coercive and would not represent real clinical conditions. No ethics committee would approve such an intervention without extensive prior work.” And, under section Ethics: “The ethical thresholds to test a coercive intervention in involuntary patients are very high. We are well aware that no ethical board would approve mechanical restraint as a new intervention if it were not common practice for centuries. However, for any novel intervention, evidence of ethical merit and adequate safety is required before real-world trials can begin. We chose to design an experimental study on ethical grounds, avoiding the direct exposure of patients to a coercive intervention.”

Comment: Since this is a pilot study, limited sample is existing. In this study, the inclusion criterion for patients was having experienced at least one mechanical restraint which means all of participants were in relative normal status during observation, which did not consider different types of disorder patients, which may greatly limit the feasibility exploration of this method.  

Author’s reply: As explained above, this is not a pilot study. In the results section, 1st par., we added the following sentence with respect to diagnoses: “Three patients suffered from schizophrenia, two from a bipolar disorder and one from complex posttraumatic stress disorder.”

Comment: Although in terms of human dignity in this study, all participants rated it positively compared to mechanical restraint, there are some potential problems in Heavy bag: for transfer from one place to another; for robust, overt violent behaviour and agitating patients, a 30kg bag is not enough to secure them well, while for petite patients, a 30kg bag is a big burden; in addition, bag may fall on the patient; and other potential dangers, which may limit its application into clinic settings.

Author’s reply: Yes, these were the concerns of professionals as mentioned in the results section. But there is also mentioned: “But all conceded that this risk could be minimised by use of a second bag in a row or at the other wrist.” To clarify this, we picked this aspect up for the discussion and added the following: “The main safety concern was whether a 30 kg bag would be sufficient to control a strong, agitated and violent patient. However, all participants agreed that the bags are a flexible tool that can be fine-tuned to individual situations. For weaker patients, a 20 kg bag would probably be sufficient and allow more mobility. For very strong patients, two 30 kg bags in a row will be sufficient to prevent any fast movements and attacks out of reach of the free hand.”

Comment: Page 2 line 86 and 87: “experience” should be changed to “experienced” or “experiencing”.

Author’s reply: We did so.

Comment: Page 4 line 145: please cross out “„” in front of “Its”.

Author’s reply: We are sorry, we did not find that in our ms, also not with the seeking tool (our paper version provided by the Editorial Office does not have line numbers).

Comments on the Quality of English Language:

I think the English expression in the article is standardized and accurate, but some details still need to be modified.

Author’s reply: We did the changes mentioned above, a native speaker did not mention further language errors.

Reviewer 2 Report

Comments and Suggestions for Authors

Dear authors!

I can see that the theme is familiar with you and subject is interesting and valuable. I suggest that you open little more about mechanicat restraint, what does it mean and what does mean seclusion. As you write, there is differences in legislation and practices between countries. In some countries, a nurse must be present by the patient's side if the patient is restrained to the bed with belts. I hope you to open define the concepts and open what does those new methods mean to the legislation in the introduction and also in the discussion.

Author Response

Dear Reviewer 2,

Comment: I can see that the theme is familiar with you and subject is interesting and valuable. I suggest that you open little more about mechanical restraint, what does it mean and what does mean seclusion. As you write, there is differences in legislation and practices between countries. In some countries, a nurse must be present by the patient's side if the patient is restrained to the bed with belts. I hope you to open define the concepts and open what does those new methods mean to the legislation in the introduction and also in the discussion.

Author’s reply: Thank you for the comments. There are now thousands of papers on mechanical restraint and seclusion so that we had viewed these terms as well-known. But we acknowledge that some explanation is certainly helpful in a more general journal. We added the following in the 1st par. of the introduction: “Mechanical restraint means restraining patients by use of belts to their bed at four (wrists, ankles), five (including abdomen) or even seven points (including head and shoulders) to achieve immobilisation and avoid harm to self and others.” And: “Seclusion means that the patient is placed in a locked room with very little furniture for security reasons, often just a mattress on the floor. Staff remain outside the seclusion room, maintaining contact through a window, sometimes watching on a video screen.” Yes, legislation will be important at some point, but we think that we are still far from legislation in this stage where the method has not yet been applied yet outside this experimental setting. We added the following sentence at the end of the discussion: “National legislation regarding the use of coercive measures must be taken into account.” But in our opinion it is far too early to go deeper in this aspect.

Reviewer 3 Report

Comments and Suggestions for Authors

Dear authors,

It's important to reiterate that the methodological issues are the main barrier preventing this pilot study from achieving scientific value. These issues, along with the motivation behind the novel method, need to be addressed.

As the methodological;

The interview's scientific description, the pilot study's requirements, the precautions taken to prevent the pilot process's effects on patients, the presentation of results, ethical and legal dimensions, the physical impact of this novel method on patients' bodies, and patient safety dimensions...

Most importantly, even though you mentioned this in your discussion, it seems that you did not follow the bundles in restraint prevention in psychiatric settings that have been proven successful but need to be worked on further. 

Kind regards.

Author Response

Dear Reviewer 3,

Comment: It's important to reiterate that the methodological issues are the main barrier preventing this pilot study from achieving scientific value. These issues, along with the motivation behind the novel method, need to be addressed.

Author’s reply: We are sorry, we discussed that, but we do not understand what you mean. The motiviation has been addressed quite clearly in the paper: To develop a method that is superior to traditional mechanical restraint under aspects of human dignity.

Comment: As the methodological; The interview's scientific description, the pilot study's requirements, the precautions taken to prevent the pilot process's effects on patients, the presentation of results, ethical and legal dimensions, the physical impact of this novel method on patients' bodies, and patient safety dimensions...

Author’s reply: Again, we are sorry, we did not understand what you mean in detail.

Comment: Most importantly, even though you mentioned this in your discussion, it seems that you did not follow the bundles in restraint prevention in psychiatric settings that have been proven successful but need to be worked on further.

Author’s reply: We are sorry, this must be a misunderstanding. We have published a review paper on measures to prevent coercion, a national guideline for the prevention of coercion, and a RCT on the implementation of this guideline in clinical practice (all not referenced in this paper). So we are really very well aware about “the bundles of restraint prevention”. But there is clear evidence that all these measures can reduce but not eliminate coercion, among others evidenced by hundreds of thousands of these measures carried out worldwide, also in well-equipped psychiatric units. Our discussion section addresses comprehensively this important difference between prevention and alternatives. All this is not really relevant here, because we did not apply coercive measures in patients but interviewed patients who had experienced such  measures previously.

Reviewer 4 Report

Comments and Suggestions for Authors

The authors struggle not to call the use of the bag a type of mechanical restrain. But it is. So, it should be labeled appropriately. Why is it an alternative to what it is (mechanical restrain)?  What is only different is the device. 

Unfortunately, I tried but could not watch the video but I can picture the scene from the authors' description. The volunteered patients should have experienced the restrain in real-time to demonstrate some "experiential knowledge" during the interview.

Comments on the Quality of English Language

It is good, only minor edit required. 

Author Response

Dear Reviewer 4,

Comment: The authors struggle not to call the use of the bag a type of mechanical restrain. But it is. So, it should be labeled appropriately. Why is it an alternative to what it is (mechanical restrain)?  What is only different is the device.

Author’s reply: Thank you for the comment. Of course you are right. We acknowledge the need to clarify that. We added the following at the begin of the discussion section: “Strictly spoken, it is also a mechanical restriction of mobility and therefore a kind of mechanical restraint. However, compared to traditional mechanical restraint, it is more flexible, allowing the patient more mobility, allowing to use the toilet and to eat and drink in a seated position and to talk to a present staff member in an upright position.”

Comment: Unfortunately, I tried but could not watch the video but I can picture the scene from the authors' description. The volunteered patients should have experienced the restrain in real-time to demonstrate some "experiential knowledge" during the interview.

Author’s reply: Oh what a pity. The video is an important part. We checked it again and it seems that it needs rather good internet connectivity. We could create a slightly changed format that works also with poorer internet connection (link changed in the ms): https://www.uniklinik-ulm.de/fileadmin/default/Kliniken/Psychiatrie-Psychotherapie-I/Bilder/2023-05-03_Fixierung_480_UT1_6mbps_str.mp4 .

Round 2

Reviewer 3 Report

Comments and Suggestions for Authors

Dear authors,

I think that the method is not innovative and does not have the necessary measurements/scientific content to be innovative.

Regards.